# Identification and Characterization of Static Craniofacial Defects in Pre-Metamorphic *Xenopus laevis* Tadpoles

**DOI:** 10.3390/jdb13030026

**Published:** 2025-07-25

**Authors:** Emilie Jones, Jay Miguel Fonticella, Kelly A. McLaughlin

**Affiliations:** Department of Biology, Tufts University, Medford, MA 02155, USA; emilie.jones@tufts.edu (E.J.); jfonticella@fas.harvard.edu (J.M.F.)

**Keywords:** tissue remodeling, craniofacial, nerves, metamorphosis, skeletal muscle, congenital birth defects, repair, vertebrate

## Abstract

Craniofacial development is a complex, highly conserved process involving multiple tissue types and molecular pathways, with perturbations resulting in congenital defects that often require invasive surgical interventions to correct. Remarkably, some species, such as *Xenopus laevis*, can correct some craniofacial abnormalities during pre-metamorphic stages through thyroid hormone-independent mechanisms. However, the full scope of factors mediating remodeling initiation and coordination remain unclear. This study explores the differential remodeling responses of craniofacial defects by comparing the effects of two pharmacological agents, thioridazine-hydrochloride (thio) and ivermectin (IVM), on craniofacial morphology in *X. laevis*. Thio-exposure reliably induces a craniofacial defect that can remodel in pre-metamorphic animals, while IVM induces a permanent, non-correcting phenotype. We examined developmental changes from feeding stages to hindlimb bud stages and mapped the effects of each agent on the patterning of craniofacial tissue types including: cartilage, muscle, and nerves. Our findings reveal that thio-induced craniofacial defects exhibit significant consistent remodeling, particularly in muscle, with gene expression analysis revealing upregulation of key remodeling genes, *matrix metalloproteinases 1* and *13*, as well as their regulator, *prolactin.2*. In contrast, IVM-induced defects show no significant remodeling, highlighting the importance of specific molecular and cellular factors in pre-metamorphic craniofacial correction. Additionally, unique neuronal profiles suggest a previously underappreciated role for the nervous system in tissue remodeling. This study provides novel insights into the molecular and cellular mechanisms underlying craniofacial defect remodeling and lays the groundwork for future investigations into tissue repair in vertebrates.

## 1. Introduction

Craniofacial development is a highly conserved, coordinated process that involves interactions between numerous regulatory networks and tissue types. This high level of complexity underlies an increased likelihood of developing a permanent craniofacial defect. Accordingly, the Centers for Disease Control and Prevention (CDC) estimates 1 in every 33 babies born in the United States will experience some form of congenital birth defect, with many of these defects classified as orofacial [1]. Currently, the treatments for these defects in humans rely on costly and invasive surgical interventions [2,3]. Though mammals generally retain their fetal phenotypes, defects included, as they grow and develop, some species are able to undergo tissue remodeling during metamorphosis to create their adult phenotypes [4,5,6,7,8].

Some species, such as the amphibian *Xenopus laevis*, take this process a step further. In addition to traditional metamorphic remodeling events, previous studies in *Xenopus laevis* have demonstrated that these animals also have a unique and robust ability to remodel mispatterned craniofacial defects during pre-metamorphic stages through a thyroid hormone-independent mechanism [9]. Although this research revealed important aspects of tissue remodeling, it remained unclear what influences allowed these animals to: (1) detect a defect in patterning and (2) promote the remodeling response. In addition, the earlier model for remodeling of malformed craniofacial tissues assumed that all defects, barring the complete loss of structures, could initiate tissue remodeling mechanisms. However, this current research has identified a novel, non-correcting craniofacial defect phenotype that provides the opportunity to directly compare the environment of a remodeling and non-remodeling phenotype during the same stages of development.

To achieve this comparison, we used a pharmacological agent, thioridazine-hydrochloride (thio), to induce a reliably remodeled craniofacial phenotype. To model non-corrective phenotypes, the anthelmintic, ivermectin (IVM), was used to induce craniofacial defects. Though the primary mechanism of action for these two compounds is different, both result in the perturbation of overall facial morphology during development, allowing us to track any correction events in pre-metamorphic animals. These teratogens induce visible defects by stage NF 45 [10], the start of the feeding stages, and can be tracked over the pre-metamorphic period until the hindlimb bud stages. These stages of development have been demonstrated to be highly plastic, with a range of craniofacial defects showing partial to complete remodeling by stage NF 49, regardless of the inducing agent [9,11].

The processes responsible for patterning craniofacial structures are generally conserved across most vertebrate species, as are the cell and tissue types involved in building the overall head and face [12,13,14,15,16]. When considering potential targets for phenotype analysis, we identified three of the main contributors to craniofacial form and function: cartilage and musculature, needed for structure and movement support, and nerves, primarily responsible for sensation and motor control [17,18,19,20,21]. Proper synchronization and organization across tissue types is vital for the proper development of the head and neck, often with each distinct structure influencing the patterning and potentially the repair of related tissue [22,23,24,25]. By mapping the individual morphological phenotypes of each tissue type in our defect groups, we further reveal the independent development and potential disruptions of these key structures and how they may play a role in determining the physical limitations of remodeling capabilities.

Although the ability to remodel some malformed structures has been previously described, to date, the underlying mechanisms involved in the pre-metamorphic remodeling process are not well understood [9,11]. Prior research revealed that animals that were able to remodel malformed craniofacial structures during early development displayed gene expression profiles distinct from both traditional metamorphic remodeling, typically controlled by fluctuations in thyroid hormone, as well as thyroid hormone (T3) induced precocious remodeling [9,26,27,28,29,30]. This research led us to investigate potential remodeling target genes such as *prolactin.2*, *mmp1*, and *mmp13*, as they have been shown to play critical roles in tissue remodeling and repair across development [31,32,33,34,35,36,37,38]. Prolactin has been implicated in modulating collagenases necessary for both metamorphic and embryonic tissue remodeling responses [35,36]. Comparison of expression patterns in these target genes between actively remodeling versus static defect groups will better elucidate the mechanisms underlying correction.

Understanding both the physical and molecular profiles of remodeling versus static, uncorrected defects is vital to demystifying the underlying requirement for pre-metamorphic correction in vertebrates. By examining and mapping the primary tissue structures of the face, we confirmed that IVM-induced defects show no significant signs of remodeling. From our thio-treated groups, we observed correction across both cartilage and muscles, with musculature showing heightened plasticity during these stages. Gene expression profiles mirrored previous work with a slight upregulation in *prl.2* as well *mmp1* and *mmp13* observed in only actively remodeling permissive treatment groups. Most significantly, we found very distinct neuronal profiles across both the remodeling and non-remodeling defect groups, suggesting the nervous system may play a more active role in juvenile remodeling than previously considered.

## 2. Materials and Methods

### 2.1. Animal Husbandry

All animal care was conducted under the guidance of the International Animal Care and Use Committee (IACUC) in accordance with protocol number M2022-92. Adult animals were housed at 18 °C under a 12/12 light–dark cycle. Oocytes were manually externally fertilized and reared in 0.1× Marc’s Modified Ringer’s solution (MMR; NaCl, KCl, CaCl, MgCl2, HEPES, EDTA), pH 7.8, from one cell to stage 49 according to standard protocols [39]. Embryos and tadpoles were housed at 14–23 °C with media changes 3 times a week. After stage 45, all tadpoles were fed 4 times a week with Sera Micron powdered growth food reconstituted in MMR. Animals were staged according to Nieuwkoop and Faber [10].

### 2.2. Pharmacological Exposures

Ivermectin (Tocris) was stored as a 10 mM stock reconstituted in dimethyl sulfoxide (DMSO). Embryos were exposed to 1 uM ivermectin in 0.1× MMR from NF stages 9 to 26; at stage 26, tadpoles were moved to fresh 0.1× MMR. Thioridazine-HCl (Sigma-Aldrich, St. Louis, MO, USA) was reconstituted in deionized water and stored at 1 mM. Embryos were exposed to 90 uM thioridazine in 0.1× MMR from NF stages 12 to 26, after which they were moved to fresh 0.1× MMR. Both drug stocks were stored in single-use aliquots at −20 °C and never refrozen. All animals were reared at 14 °C until drug treatments, when they were moved to 18 °C until fixing. Drug concentrations were chosen through dose–response experiments, ensuring consistent defects without significant death.

### 2.3. Geometric Morphometric Analysis

Stage 45 and 48 tadpoles were temporarily immobilized with 0.5% tricaine and imaged both ventrally and dorsally using a Nikon SMZ1500 dissection microscope with Spot Insight Color camera and Spot Image Solution Software. The StereoMorph R package, version 1.4, was used to assign landmarks to each image, and analyses on relative distances and angles of each structure were performed using Microsoft Excel [40]. The dorsal and ventral landmarks for cartilage assessment were assigned according to the provided schematic in Figure 1F. Landmarks for musculature assessment are outlined in Figure 2F. Landmarks for both tissue types and their target morphologies are outlined in Table 1 and Table 2. Correction was determined by statistical comparison of each treatment to its corresponding control group for each landmark parameter. A group was determined to be corrected when they were no longer significantly different from the control averages. Partial remodeling refers to groups that showed a reduction in significance level from stage 45 to stage 49.

### 2.4. Whole-Mount Immunohistochemistry

Tadpoles were euthanized using 0.5% tricaine (Sigma-Aldrich, St. Louis, MO, USA) and fixed for 1 h in a 1XMEMFA (MOPS, EGTA, MgSO_4_, Formaldehyde; Fisher Scientific, Hampton, NH, USA) solution. They were then rinsed once and washed 4 times for 15 min with 1× phosphate-buffered saline (PBS; NaCl, KCL, Na2HPO4, KH2PO4; Fisher Scientific, Hampton, NH, USA). Tadpoles were incubated in block, consisting of 20% inactivated goat serum in PBS + triton X-100 (PBTr, Sigma-Aldrich, St. Louis, MO, USA), for 1 h at room temperature (RT) then incubated in primary antibody (nerves visualized with anti-acetylated tubulin, Sigma-Aldrich, St. Louis, MO T7451-25UL; muscles visualized with 12/101, DSHB, Iowa City, IA, USA) diluted 1:1000 in block overnight at 4 °C rocking. The samples were then washed 4 times with PBTr for 1 h each rotating at RT. Samples were then blocked for 1 h at RT before incubating with secondary antibody (Alexa-Fluor-555, Invitrogen, Waltham, MA, USA, at 1:300 in blocking solution) overnight at 4 °C. Secondary was then washed out with PBTr and the samples were imaged using a Leica Thunderstorm scope on Leica Las X software version 3.7.2.22383.

### 2.5. Alcian Blue Staining

Tadpoles were fixed overnight in ethanol (ETOH; Fisher Chemical, Branford, CT, USA) then transferred to 0.01% alcian blue (0.1 mg/mL alcian blue, Fisher Scientific, Hampton, NH, USA, in a 1:4 acetic acid, Fisher Chemical, Branford, CT, USA, to ETOH solution). Tadpoles were stained rocking for two hours, rinsed once with 70% ETOH then carried through two 30 min 1% hydrochloric acid (HCl, Fisher Chemical, Branford, CT, USA) washes. Tadpoles were left in HCl overnight rocking then rinsed three times with 95% ETOH. They were then gradually rehydrated into PBS and fixed 1× MEMFA as described above. Tadpoles were then bleached in a 30% hydrogen peroxide solution (Fisher Chemical, Branford, CT, USA) followed by maceration in 2% potassium hydroxide. Once external tissue was cleared, they were gradually moved to 80% glycerol for storage.

### 2.6. Image Adjustments

Immuno and alcian blue-stained images were adjusted for increased visibility using Microsoft PowerPoint version 16.96.1. Alcian blue cartilage-stained images were brought up 36% for brightness. Musculature image brightness was increased by 41%. Nerve stain images were adjusted + 50% for brightness and + 25% for contrast. Saturation for all nerve images was set to 0 to present black and white images. Adjusted images were not used for any quantification and have been provided to their editor for transparency. Original nerve stain images used for quantification can be seen in Appendix A.

### 2.7. Real-Time Quantitative PCR (RT-qPCR)

Samples for qPCR consisted of stages 45 and 48 *Xenopus laevis* heads, structures anterior to the gut. Batches of five individual tadpoles were pooled following RNA extractions for a total of 10 per group for cDNA synthesis. RNA extractions were performed by phase separation according to the TRIzol (Thermo Fisher, Waltham, MA, USA) protocol with 5-bromo-4-chloro-3-indolyl phosphate (BCIP, Sigma-Aldrich, St. Louis, MO, USA) substituted for chloroform. cDNA synthesis was carried out with M-MuLV Reverse Transcriptase according to New England Biolab’s first strand cDNA synthesis quick protocol (NEB #M0253, Ipswich, MA, USA) using d(T)23 VN oligo primers. Concentrations for both RNA and cDNA were assessed via nanodrop200 (Thermo Fisher, Waltham, MA, USA). Applied Biosystem’s PowerUP SYBR Green Mastermix (Applied Biosystems, Woburn, MA, USA) was used for RT-qPCR reactions, and the samples were run on an Applied Biosystems 7300 Real-Time PCR machine. Primer sequences can be found in Appendix A.

### 2.8. Statistical Analysis

Landmarks were analyzed by non-parametric *t*-test (Mann–Whitney U-tests) to assess significance in geometric morphometric analysis. Outliers for each experiment were removed using the ROUT method (Q = 1%). Quantitative PCR results were set to a biological significance level of 1× fold change with inter-treatment significance determined with Welch’s *t*-tests. All statistics were performed using GraphPad Prism 9.

### 2.9. Behavioral Scoring

At NF stages 45 and 48, tadpoles were moved to observation dishes and were allowed to acclimate for five minutes. Tadpoles were then imaged for three minutes using an iPhone mounted to a stereo microscope. Videos were then scored in slow motion for abnormal movements (side swimming, twitching, spinning). Normal swimming was considered dish circling, dish crosses, or collision response movements. Tadpoles were observed in 100 mm × 15 mm petri plates with 10 individuals per dish.

## 3. Results

### 3.1. Ivermectin-Induced Craniofacial Defects Show Limited Remodeling Capabilities of Major Landmark Structures Through Pre-Metamorphic Stages

To better understand the physical parameters of craniofacial remodeling, geometric morphometric mapping was used to track the changes in shape evolution of major facial features in pre-metamorphic tadpoles that could either remodel altered cranial facial structures or display limited remodeling ability. These landmarks and their target morphologies are outlined in Table 1. The previously described craniofacial defect-inducing teratogen, thioridazine hydrochloride (thio), was used as an example of a well-characterized, self-correcting condition [9]. Results from the thioridazine hydrochloride-treated tadpoles were compared to exposure to a novel, non-remodeling pharmacological treatment, ivermectin (IVM). Defects were assessed both dorsally, observing changes in eye shape as well as mouth angles, and ventrally, examining size and positional changes of major cartilage groups (Figure 1). Representative dorsal images for these defects are shown in Figure 1A. As reported previously, thioridazine hydrochloride-treated *X. laevis* tadpoles exhibited remarkable capacities to correct even major craniofacial defects in both soft tissues, such as the eye, as well as cartilaginous structures in thioridazine-treated animals [9]. In sharp contrast, defects induced by ivermectin IVM treatments resulted in malformations that either persisted or worsened during later pre-metamorphic stages. Schematics for morphometric landmarks can be found in Figure 1C.

Though normal morphology was observed at early NF stages (NF 45), eye morphology, measured by taking the ratio of the length and width of the left eye, in the ivermectin-treatment group showed significant abnormalities by NF stage 49 (Figure 1D). In contrast, thioridazine-treated animals showed significantly abnormal ratios at stage 45 (Mann–Whitney U-test, *p* < 0.0001), which were resolved to a normal phenotype by NF stage 49 (Figure 1D). Interestingly, when observing the remodeling trends of the individual ratio components, there seems to be an overall stronger corrective trend for lengthening rather than widening of the shape, resulting in more ovular eyes (Appendix A). While we observe a correction in both metrics in thio-treated tadpoles, the eye widths, but not lengths, remain statistically distinct compared to control groups at stage 49 (Mann–Whitney U-test, vs. *p* < 0.01) (Appendix A). Even in ivermectin treatments, where we see persistent defects in overall eye shapes, we observed a trend towards normal lengthening, suggesting a potential planar preference in this tissue during correction (Appendix A).

To characterize the overall shape of the jaw, the angle of the tadpole mouth was measured using the forebrain and mouth corner as landmarks at NF stages 45 and 49 (Figure 1E). At stage 45, both treatment groups showed significant (Mann–Whitney U-test, *p* < 0.0001) deviations from normal head morphology observed in the untreated control animals. Ivermectin-induced defects became more pronounced by stage 49, with a narrowing of the jaw resulting in a reduced mouth angle. Thioridazine-induced defects were shown to normalize by stage 49 with angles comparable to control untreated groups (Figure 1E). These results support that our remodeling and non-remodeling groups are following the predicted trends in correction, but they do not allow us to visualize what individual craniofacial structures are undergoing remodeling during these stages. To better understand how these overall morphological abnormalities are achieved, we looked at three prominent facial structures: the Meckel’s, quadrate, and ceratohyal cartilages.

The horizontal spread of Meckel’s and quadrate cartilages, as well as the angle of the left ceratohyal cartilage, were examined (Figure 1F–H). In tadpoles, the Meckel’s and quadrate cartilages are largely responsible for the shape and size of the lower jaw [41]. Thus, to best capture remodeling in this area, changes in spread between both these groups were measured, and the average distance was quantified for each. The quadrate results showed severe defects in ivermectin-treated groups at NF stage 45, resulting in significantly condensed cartilage (Mann–Whitney U-test, *p* < 0.0001) (Figure 1G). These effects persisted throughout the pre-metamorphic period, with aberrant phenotypes worsening by stage 49 (Figure 1B). Though imperfect (Mann–Whitney U-test, *p* < 0.01), we did observe some normalization of thioridazine-induced defects in this structure. In general, the thioridazine-treated groups displayed milder defects in this area at stage 45, with some minor abnormalities persisting into stage 49 (Figure 1B).

A similar trend was observed when quantifying the Meckel’s cartilage spread. Though both groups displayed condensed lower jaws, this effect was most obvious in the ivermectin-induced defects (Figure 1A). Thioridazine-treated groups as a whole trended toward a more normal morphology, though still statistically distinct from untreated groups (Mann–Whitney U-test, *p* < 0.05). In contrast, we observed a worsening of Meckel outcomes in every individual in the ivermectin-treatment group (Figure 1F).

The second major cartilage group responsible for contributing to the overall face shape is the ceratohyal cartilage. Correction was quantified using landmarks at each corner of the left structure, assessing the changes in angle degree from stages 45 to 49 (Figure 1C). Thio and IVM displayed similar levels of mispatterning at stage 45 (Mann–Whitney U-test *p* < 0.01) with thio-induced defects resolving by stage 49 (Mann–Whitney U-test *p* > 0.05) and IVM-induced defects becoming more pronounced by stage 49 (Mann–Whitney U-test *p* < 0.0001) (Figure 1H). It is worth noting that the ceratohyal inner length (Figure 1C, light green) for thioridazine-treated groups was not completely corrected (Mann–Whitney U-test *p* < 0.05) by stage 49, though the overall length was drastically improved (Appendix A). The ceratohyal and jaw cartilages develop in close proximity to one another and thus may exert physical force during jaw formation, influencing the remodeling capacities of one another [42,43,44]. Results from analysis of both of these structures suggest a potential correlation of condensed jaw cartilages and the inhibition of normalization of ceratohyal interior length, either due to improper cellular coordination, as proposed by Rose et al., or biophysical influences via connective tissues [45].

Combined, these results reveal important differences between static (non-remodeling) phenotypes resulting from ivermectin exposure and the remodeling abilities of thioridazine-induced defects. Quantifying the correction of individual cartilage groups has allowed us to better understand the scope of correction at this level to parse potential structure-specific limits to remodeling.

### 3.2. Major Musculature Defects in Ivermectin-Treatment Groups Fail to Show Evidence of Remodeling Through Pre-Metamorphic Periods

Correction of the abnormal structures in the tadpole’s face includes coordination of more than just cartilage groups. Muscle is a highly plastic tissue type that plays an important role in shaping the face throughout development. Interestingly, musculature has been shown to remodel in response to stimuli such as environmental changes, exercise, and injury [46,47,48]. To better understand how the remodeling capacities of this tissue type compare to those of the major cartilage structures, geometric morphometric mapping on immunostained animals was used to examine normalization of defects at early and late pre-metamorphic stages (Figure 2). Schematics and images for representative muscle phenotypes can be found in Figure 2G, and morphometric landmarks are outlined in Figure 2F as well as Table 2.

The intermandibularis and orbithyoideus muscle groups correspond with the quadrate and Meckel’s cartilages, respectively. Both ivermectin and thioridazine treatment groups showed significant (Mann–Whitney U-test, *p* < 0.0001) orbitohyoideus defects at stage 45, but by stage 49, the thioridazine-treated group had normalized to a control phenotype while ivermectin-induced defects remained significantly distinct from control untreated groups (Figure 2A). In contrast, defects in the intermandibularis only showed modest remodeling in thioridazine-treatment groups and an overall worsening of defects in the ivermectin-treated animals, with both displaying severely condensed musculature in this area (Figure 2B). These results mirror what we observed in the cartilage analysis, with more complete remodeling occurring in more rostral jaw structures.

To measure lengthening or vertical spread of the head, we also analyzed the geniohyoideus and interhyoideus cartilage lengths as landmarks for morphometric analysis. The interhyoideus muscle correlates positionally with the ceratohyal cartilage. This thick band of muscle was severely condensed in the ivermectin-treated group, showing no signs of remodeling into stage 49 (Figure 2E). This aligns with our results presented from the ivermectin-induced ceratohyal defects, as the vertical condensation of both structures showed persistent mispatterning throughout pre-metamorphic stages. Interestingly, though this pattern was consistent for ivermectin-induced defects, thioridazine-induced defects displayed significant muscle remodeling in this area despite their failure to correct corresponding cartilage defects. Similarly, in ivermectin treatment groups, the geniohyoideus, a component of the lower jaw, shows a stark increase in mispatterning throughout pre-metamorphic stages, while thioridazine-induced defects trended toward a normalized phenotype (Figure 2D). These data demonstrate that facial tissue types can remodel at independent rates from one another, with muscle tissues more readily remodeling when compared to events occurring in the facial cartilages.

### 3.3. Remodeling Treatment Groups Display Distinct Peripheral Nerve Phenotypes Compared to Control Groups

In addition to the remodeling capacity differences observed in ivermectin- and thioridazine-treated animals, both treatment groups showed distinct behavioral profiles. As ivermectin is a known anesthetic, we unsurprisingly observed an overall reduction in movement throughout the pre-metamorphic period. By contrast, thioridazine treatments increased early twitching behaviors as well as disrupted normal, coordinated swimming patterns (Appendix A). To better assess a potential underlying mechanism of these abnormal behaviors, the neuronal marker acetylated tubulin was used to characterize the patterning of the peripheral nervous system.

Results from fluorescent immunochemistry revealed a severe reduction in the overall number of visible peripheral nerves in both the head and tails of ivermectin-treated individuals. Looking specifically at the craniofacial nerves, by stage 45, we observed a reduction in acetylated tubulin and branching patterning around cranial facial nerves I, II, III, V, and VIII, as well as a loss of craniofacial nerves IX and X (Figure 3A). Similarly, ivermectin groups at these early stages showed a loss of all three tail nerve types: commissural fibers, longitudinal fibers, and internal neuropil (Figure 3B). By stage 49, there was some recovery of craniofacial nerve branching in ivermectin-treated animals, particularly in the optic nerve surrounding area, though there remained an overall reduction in nerves IX and X (Figure 3A). Stage 49 tails did show commissural fiber growth along the somites, though these structures appeared underdeveloped compared to control animals. Though there was some mild branching from the commissural fibers, there remained a reduction in internal neuropils and a loss of longitudinal fiber growth.

In contrast, in thioridazine-treated groups, the peripheral nervous system was characterized by increased branching and disorganization in both the craniofacial region and the tail (Figure 3A,B). This phenotype was most prevalent in stage 45 tadpoles but remained observable through stage 49. Although stage 49 thioridazine-treated groups showed a trend toward normalization of craniofacial nerves, the disorganization of peripheral nerves of the tail appeared to increase through pre-metamorphic stages (Figure 3B). Stage 45 tadpoles showed normal commissural fiber and internal neuropil growth with an increase in longitudinal fibers, while stage 49 animals showed increased branching and mispatterning of the commissural fibers along with an increase in longitudinal fibers.

Quantification of the craniofacial nerve profiles for the two groups, as assessed by ImageJ (version 1.54g) analysis, revealed a linear trend to a normalized phenotype in the ivermectin- but not thioridazine-treated groups by stage 49 (Appendix A). Additionally, qualitative scoring of these phenotypes revealed that 100 percent of ivermectin-treated tadpoles showed some visual evidence of hypo-innervation by stage 45, with the majority still displaying obvious abnormalities by stage 49 (Appendix A). Thioridazine-treated animals had nearly 100 percent hyper-innervation at stage 45, with nearly 50 percent recovering a control phenotype by stage 49 (Appendix A). Though ivermectin-treated groups seem to at least partially recover their behavioral and nerve signatures, their craniofacial structures remained severely mispatterned.

### 3.4. Remodeling and Non-Remodeling Treatment Groups Display Distinct Expression of Remodeling-Associated Genes During Pre-Metamorphic Stages

Previous research has shown a likely role for prolactin hormone 2, a modulator of key matric metalloproteinases, in pre-metamorphic remodeling [9]. Prolactin has also been implicated in neuroprotection and neurogenesis, potentially playing a role in our remodeling tadpoles’ hyper-innervated phenotype [49,50,51]. Our RT-qPCR data revealed a slight upregulation of *prl.2*, which becomes more pronounced by stage 48 in thioridazine-treated remodeling tadpoles (Figure 4A). In contrast, ivermectin-treatment groups displayed a very minimal upregulation at stage 45 and a slight downregulation by stage 48.

Interestingly, ivermectin treatment groups also showed a slight upregulation (about a 1.4 and 1.3 Log2 fold change, respectively) of both thyroid hormone receptors alpha (*thra)* and beta (*thrb*), genes associated with typical tadpole metamorphosis, at stage 45, though both groups exhibited a decrease in expression by stage 48 (Figure 4A). It is important to note that the expression for these genes did not extend beyond the biologically significant range, though the trends were consistent between trials. Two prolactin-mediated matric metalloproteinases, *mmp1* and *mmp13*, were also observed to be upregulated in late-stage remodeling groups (Figure 4B). Thioridazine-treated groups exhibited a modest increase in *mmp13* expression (1.5 Log2 fold change) with a much more robust up-regulation of *mmp1* (17.4 Log2 fold change) by stage 48. In contrast, ivermectin treatment groups showed only a slight increase in *mmp1* expression along with a downregulation of *mmp13* by stage 48 (Figure 4B). Between these two genes, only thioridazine stage 48 *mmp1* can be considered biologically significant, though there was very little variation in expression trends between biological replicates.

## 4. Discussion

To better understand potential physical parameters underlying remodeling processes, multi-tissue morphology was assessed using a combination of immunohistological staining and morphometric mapping. Additionally, previously identified remodeling-related pathways were assessed for changes in gene expression using RT-qPCR. These findings provide a new and exciting layer to the story of *Xenopus* tissue remodeling. Gene expression profiles across remodeling and non-remodeling treatment groups reveal a likely role for the hormone prolactin in tissue remodeling, rather than a more traditional metamorphic, thyroid hormone mode of control (Figure 4). The drastic upregulation of *mmp1* in later-stage remodeling tadpoles suggests a prolactin-mediated remodeling response, likely through increased cell proliferation and migration [35,52,53]. The slight upregulation of *mmp13* also suggests a potential role for collagen degradation in the cartilage remodeling process [38,54,55]. Both prolactin and MMPs are known to play critical roles in tissue remodeling across development, making them ideal candidates for potential roles in pre-metamorphic correction events [31,32,33,34,35,36,37]. On a morphological level, extended mapping showed static or worsening defects in cartilage, muscle, and peripheral nerves extending into stage 49 in ivermectin-treated tadpoles compared to the near full remodeling of thioridazine-induced defects. Direct comparisons of specific cartilage and muscle groups used to assess each structure’s innate remodeling capacities revealed that even in remodeling permissive sites, those that showed reliable remodeling in thioridazine groups, ivermectin-induced defects remained mispatterned throughout the pre-metamorphic period. This phenomenon was especially pronounced in the major facial muscles of these groups (Figure 2G).

Across species, muscles have been shown to be highly plastic and readily remodel in response to mechanical forces within the affected area [46,47]. We did see evidence of this plasticity in our thioridazine “remodeling” treatment, as they showed remarkable correction toward normal phenotypes in all major jaw muscle groups. As the facial muscle and cartilage structures are so interconnected, we predicted that there would be similar rates of remodeling across corresponding tissues. What we observed was actually a correction in some specific muscle groups despite a worsening of their corresponding cartilage structure in thioridazine-treated tadpoles, suggesting tissues can remodel independently from one another. We observed that some cartilage structures, particularly in the more rostral lower jaw region, show limited evidence of correction, particularly when presenting with a condensed phenotype (Figure 1). Despite these areas showing incomplete correction, the overlying musculature does retain its expected plasticity in the interhyoideus and, to a lesser extent, the intermandibularis structures in remodeling groups.

The same phenomenon was not observed in the ivermectin-induced defects, with correction of both cartilage and musculature remaining mispatterned into stage 49 (Figure 2). These data suggest that even in facial structures like musculature, which have high remodeling potential, there is a failure to recognize or initiate phenotype correction in ivermectin-treated tadpoles.

While the morphology of these specific defects likely plays some part in determining normalization outcome, one of our most interesting findings was in the distinct morphology of the peripheral nerves. Our treatment groups displayed distinct neuronal phenotypes, with remodeling groups exhibiting increased branching and disorganization of both cranial and tail nerves, and non-remodeling groups revealing a reduction in these nerve groups.

Interestingly, previous work with ivermectin has shown that treated animals develop a hyper-innervated phenotype as a result of surgical manipulations, suggesting these depolarized animals have the potential for repair but fail to activate these mechanisms in response to internal stimuli [56,57]. This repair mechanism makes sense as peripheral nerves have been implicated in size control and muscle maintenance, both during development, as well as adult tissue upkeep across species [58,59,60,61,62,63].

When considering the overall patterning of the nervous system, it is important to consider the developmental stages affected by each perturbation. Our remodeling treatment group, as well as the various remodeling phenotypes described by both Pinet and Vandenberg, primarily target neurula to tailbud stage embryos. During these early stages, the neural crest, which is largely responsible for what will become both the craniofacial skeleton and the peripheral nervous system, is undergoing specification, delamination, and migration [64,65,66,67]. In contrast, our non-remodeling treatment can be administered post-tailbud stages, suggesting a neural crest-independent mechanism of defect induction. As these two groups could be targeting unique cell or tissue groups to influence patterning, it is likely that they could respond differently to remodeling mechanisms. It is important to note that, within the early stages of development, previous research has shown that the mechanism of defect initiation is largely irrelevant in determining remodeling abilities [9,11]. Manipulations to dopamine signaling, chondrogenesis, and V-ATPase pump functioning all result in unique craniofacial phenotypes with the ability to normalize during pre-metamorphosis [9,11]. Interestingly, though we observed stark differences in remodeling rates across our compounds of interest, both treatments have been shown to play a role in reducing tumor cell growth through similar cellular processes such as cell cycle arrest as well as increased rates of cytotoxicity and autophagy [68,69,70,71,72].

Though induced during different stages of neural development, both treatments influence the peripheral nervous system patterning and function. Our remodeling defects are induced by a pharmacologic agent, thioridazine, which works as both a dopamine receptor agonist and glutamate modulator [73,74,75,76]. Both pathways have potential excitatory implications within the developing nervous system. Additionally, thioridazine has also been identified to disrupt potassium and calcium ion signaling in medulloblastoma cells. These thioridazine-induced perturbations were linked to reduced mitosis rates and abrogated metastasis of tumor cells [68]. Given the increasingly appreciated importance of bioelectric state in peripheral nervous system activity, the off-target disruption of potassium and calcium ion signaling pathways could be another contributing factor to the hyper-innervated phenotype induced by thioridazine in our *Xenopus* model [77]. In contrast, our non-remodeling defect-inducing agent, ivermectin, is also a multi-target drug wherein one effect is to act as a GABA/glutamate-gated channel ligand and results in a temporary period of paralysis. Importantly, in vertebrates, this compound acts as an anesthetic rather than a traditional paralytic, allowing for the prevention of sensory communication without disrupting normal muscle functioning [78,79,80,81].

Though the nervous system has previously been considered simply a sensory communication center, recent work has highlighted novel roles for the central and peripheral nervous systems (CNS/PNS) in early developmental patterning, regeneration, and tissue maintenance [82,83,84]. The dual roles of tissue integration and pattern development made this system an important target in our phenotype characterization.

The potential role for the nervous system is supported by our molecular findings, as prolactin not only plays a role in metamorphic remodeling, but also in neuromodulation of the peripheral nervous system [85,86,87]. Additionally, prolactin further interacts with the nervous system via dopaminergic signaling pathways, which are regulated in an oppositional manner by our two defect-inducing agents [88,89,90,91]. Future experimental goals will focus on modulating neuronal functioning in known remodeling treatment groups to assess changes in plasticity, as well as more fully characterizing the PNS profiles of similar static defects to better understand the mechanisms of defect initiation and continuation throughout pre-metamorphic stages.

## Figures and Tables

**Figure 1 jdb-13-00026-f001:**
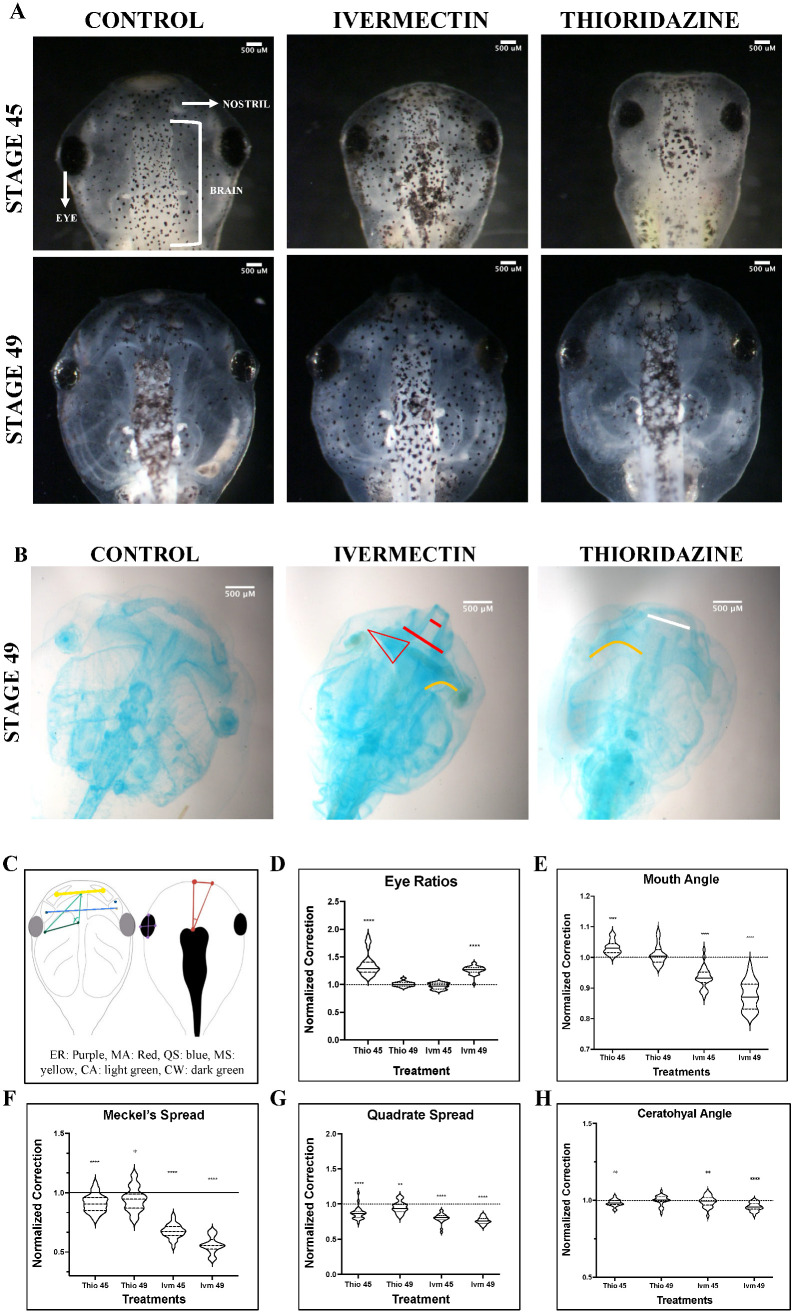
Geometric morphometric analysis of landmark correction. Morphological landmarks assessments for dorsal (**D**,**E**) and ventral (**F**–**H**) landmarks for ivermectin (IVM)- and thioridazine (Thio)-treated tadpoles were normalized to stage-matched untreated controls at stages 45 and 49. Representative brightfield images for both stages are shown in panel (**A**). Alcian blue images were not used for quantitative analysis but were provided as additional representation for remaining defects (Panel (**B**)). Schematic of geometric landmarking is shown in panel (**C**) (Eye Ratio [ER]: Purple, Mouth Angle [MA]: Red, Quadrate Spread [QS]: blue, Meckel’s Spread [MS]: yellow, Ceratohyal Angle [CA]: green, Ceratohyal Width [CW]: dark green). The white line represents persistent, but mild, defects. Red lines represent severe uncorrected defects. The orange line shows abnormal curvature of the ceratohyal cartilages. A two-tailed Mann–Whitney U-test was used for analysis. ns = *p* > 0.05, * *p* < 0.05, ** *p* < 0.01, **** *p* < 0.0001. An N of 1 is represented for each landmark with 24 to 30 animals used for each treatment group.

**Figure 2 jdb-13-00026-f002:**
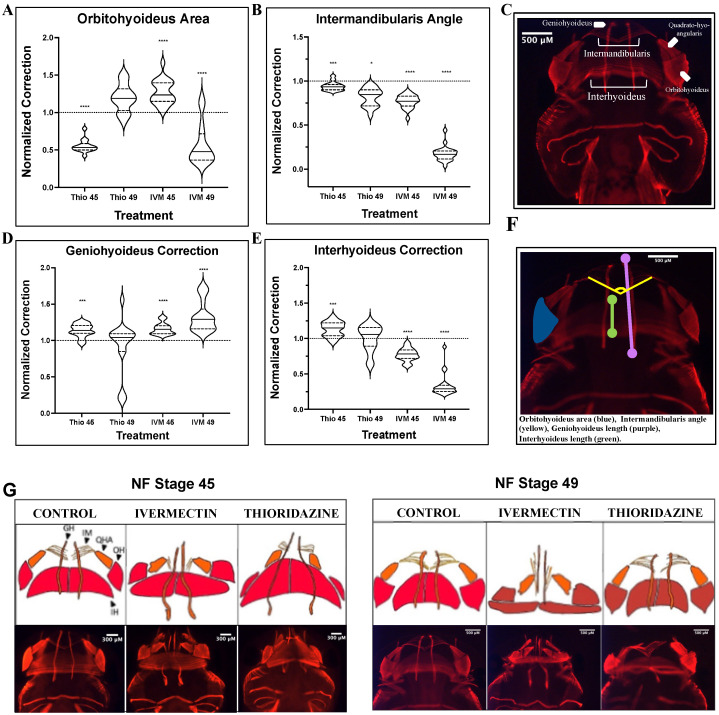
Analysis of pre-metamorphic musculature landmark correction shows persistent defects in ivermectin-treated tadpoles. 12/101 Immunohistochemistry-stained embryos stages 45 and 49 analyzed for (**A**) orbitohyoideus area, (**B**) intermandibularis angle, (**D**) geniohyoideus length, and (**E**) interhyoideus length. Schematics for muscle groups (**C**) and treatment group phenotypes (**G**), as well as the morphometric landmarks used in quantification (**F**), are provided for reference. (**F**) Landmarks used: orbitohyoideus area (blue), intermandibularis angle (yellow), geniohyoideus length (purple), interhyoideus length (green). A two-tailed Mann–Whitney U-test was used for analysis. ns = *p* > 0.05, * *p* < 0.05, *** *p* < 0.001, **** *p* < 0.0001. An N of 1 is represented for each major muscle group, with 15 to 25 animals used for each treatment group.

**Figure 3 jdb-13-00026-f003:**
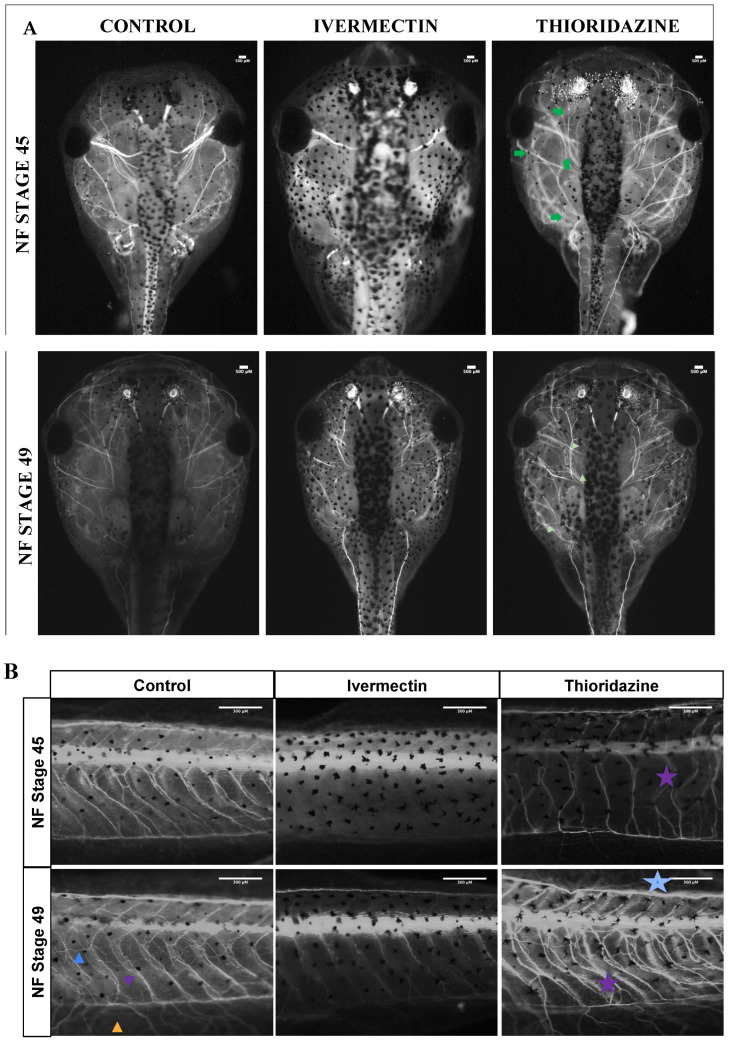
Unique innervation profiles observed between treated pre-metamorphic tadpoles and untreated controls. (**A**) Dorsal cranial images of acetylated tubulin immunohistochemistry for treatment groups at stages 45 and 49. Major cranial nerves denoted with roman numerals. Corresponding tail images for all groups are shown in panel (**B**). Increased branching is noted with green arrows. The three nerve fiber types are noted as: longitudinal (blue arrow), commissural (purple arrow), and neuropil (orange arrow). Aberrant phenotypes are highlighted with stars of corresponding colors. All data analysis was performed on unaltered images, which can be found in Appendix A.

**Figure 4 jdb-13-00026-f004:**
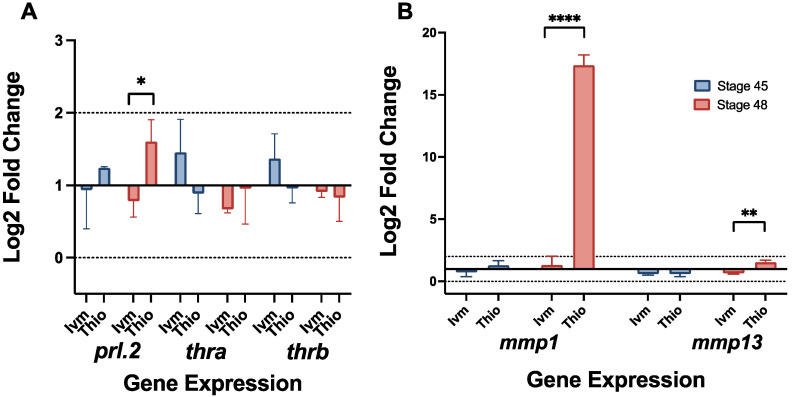
RT-qPCR reveals upregulation of remodeling-related genes prl.2 and mmp1. Gene expression profiles for (**A**) *prl.2*, *thra*, *thrb*, (**B**) *mmp1*, and *mmp13* at NF stages 45 and 49 for ivermectin- and thioridazine-treated tadpoles. Expression was calculated via the ddct method with eukaryotic elongation factor 1 (*eef1a*) as an endogenous control. A biological significance threshold was set to >1x Log2 fold change, shown by dotted lines. Error bars represent standard deviation. For all groups, *n* = 3, with each trial including cDNA from 10 pooled individual tadpoles. ns = *p* > 0.05, * *p* < 0.05, ** *p* < 0.01, **** *p* < 0.0001.

**Table 1 jdb-13-00026-t001:** Geometric Morphometric Landmarks for Cartilage Analysis.

Landmark	Orientation	Morphology Assessed
**Eye Ratios**	Dorsal	Shape of the eye
**Mouth Angle**	Dorsal	Overall width of the most anterior jaw
**Meckel’s Spread**	Ventral	Specific width of the anterior jaw structures
**Quadrate Spread**	Ventral	Specific width of the posterior jaw structures
**Ceratohyal Angle**	Ventral	The curvature and lengths of the ceratohyal cartilages

**Table 2 jdb-13-00026-t002:** Geometric Morphometric Landmarks for Muscle Analysis.

Landmark	Corresponding Cartilage	Morphology Assessed
**Orbitohyoideus Area**	Quadrate	Growth and expansion of this muscle
**Intermandibularis Angle**	Meckel’s	Widening or condensing of the anterior jaw
**Geniohyoideus Correction**	N/A	Lengthening of the overall jaw
**Interhyoideus** **Correction**	Ceratohyal	Anterior-posterior expansion of this muscle

## Data Availability

The data presented in this publication are available upon request from the corresponding author.

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
