# Peer review of "Identification and Characterization of Static Craniofacial Defects in Pre-Metamorphic *Xenopus laevis* Tadpoles"

_jdb, 2025, doi:10.3390/jdb13030026_

Round 1
Reviewer 1 Report
Comments and Suggestions for Authors
This manuscript has some really interesting and exciting data and would love to see it published. However, in its current state I don’t think it is ready for publication.
1) The first major problem is that it is primarily a descriptive study with only one gene expression experiment that starts to get to mechanism. It is missing how the remodeling occurs (does it involve production of new cells? Stem cells?) and how this is different in the two treatments. The expression of prolactin and mmps is a start, but how do these genes help to restore the size of jaw cartages for example? I think there needs to be some more experiments that link the gene expression changes and the remodeling that occurs so that you go beyond correlative and descriptive data. For example, in the Thio treated embryos does knockdown or inhibition of mmp1 prevent remodeling? What cells express mmp1 during the remodeling process? Is mmp1 involved in restoring the size of jaw cartilage or muscle or does it facilitate only specific remodeling steps? Some simple experiments maybe using an MMP inhibitor and IHC could get at these types of questions.
2) The second major problem is the readability of the manuscript.
- a) The first way to to improve readability is in the presentation of the data.
- Each figure should be set up in the same way with, first, pictures of representative embryos set up as in figure 3A. Some basic labels could be in the figures to help orient the reader (eg eyes, mouth, nostrils, brain). This could then be followed by a schematic of the measurements taken (there should be labels or a key in the figure so readers don’t need to hunt through the figure legend). Then the graphs presented.
- In terms of the graphs these need to be better explained. It is not clear what normalized correction means and what is statistically compared.
- Graphs of quantification need to be included in figure 3.
- The font size needs to be increased in figures.
- The brightness needs to be increased in fluorescent images.
- If the alcian blue labeled embryos are not used for quantitative analysis they could be added as a supplemental image.
- The brightness/contrast/saturation adjustment numbers don’t need to be added to figure legends-this number does not mean much and could be specific to the program used. Instead in the methods explain the adjustments that were made. As long as you are assessing a pattern, location, size of morphological features that does not depend on the adjustments made I think it is OK to do this for presentation purposes.
- The numbers for each experiment could be reported in the figure legends (in some cases there is a range-be precise).
- b) The second way to do this is to improve the writing of the document.
- The document could be much more concise. There is a lot of repetition. For example lines 41-43 makes a point that is then repeated in the next paragraph.
- In the entire results section, the language used is too fuzzy. There are a lot of “mildly abnormal”, “not completely corrected”, “though imperfect” , “more complete remodeling” “trend toward”, “relatively normal morphology”, “slight down regulation”. Instead, you could be precise add numbers and/or fold changes to describe the differences in the measurements and then note whether these changes are statistically different.
- Carefully consider the wording of the subheadings in the results and be precise. Eg “3.4 Remodeling and non-remodeling treatment groups…” In this section you are not assessing expression profiles but the expression of 5 specific genes and you are assessing this in thio and IVM treated embryos. Also “unique” is the wrong word choice here.
- For each measurement taken it needs to be more precisely defined either in the methods or when first mentioned in the results. For example, angle of the tadpole mouth…was measured using the forebrain and mouth corner” why would this be “mouth angle” and what does this measurement tell you about changes in craniofacial morphology? What is it a measure of? Each measurement taken should be more clearly defined.
- There are several instances of descriptions that are not accurate. For example”track the movement and shape evolution” You are not tracking movement in this work but rather the changes in landmark position or size at two different times. The structures are not “moving”. Another example” exert physical pressure” maybe force would be more accurate. Another….“cartilage group responsible for patterning the overall face shape” that suggests that the cartilage “patterns” which is not true but rather you might want to say that the cartilage contributes to the shape of the face.
- There are many grammatical errors, jargon use, inaccurate use of words and poorly constructed sentences that make this difficult to read. For example “remodeling tadpoles”, “less readily corrected”, “or initiate corrective movement”.
- Don’t abbreviate CF. There is no need since it is one word. CF most commonly stands for Cystric Fibrosis. You include a wider audience by reducing unnecessary abbreviations.
- The final major concern is there could be more depth in the approach and discussion. First, there are only 2 different compounds used and whether they represent general principles or are compound specific are unclear. Also, there could be some consideration of how these compounds work to cause developmental defects which could be really important. For example, is the chemical action reversible in each treatment? Also what are the cellular effects of each drug and could that influence the ability to later remodel the tissue. An analysis or minimally an overview of the known cellular effects (cell cycle, apoptosis, necrosis, signaling) might lead to better insight into when remodeling can and can’t occur.
As mentioned in the suggestions for authors I believe the writing of manuscript could be greatly improved.
Reviewer 2 Report
Comments and Suggestions for Authors
This study compares the effects of thioridazine hydrochloride treatment in tadpoles with those of a novel, non-remodeling pharmacological treatment, ivermectin. The authors utilized the Xenopus laevis model, an ideal model organism for such research. The study is well-written, and the experimental design is robust.
General comments:
The introduction is comprehensive and cites relevant references, most of which are more or less recent. However, some sections, such as the paragraph between lines 92-102, could be more appropriately placed in the conclusion.
The materials and methods section could benefit from additional details, such as an explanation of why the authors chose these specific concentrations. The description of the mobility/behavioral assay is entirely missing, including important information about the equipment used. Furthermore, it would be helpful to know the duration of the tadpole filming and tracking, as well as the conditions under which these observations were made. What equipment did the authors use for the experiment? How long were the tadpoles filmed and tracked, and what were the conditions during the observations?
The results section is clear, and the discussion effectively summarizes the study's findings, the figures are well-made and easy to interpret.
Specific comments:
Line 51-53 - recent research mentioned but not cited
Line 76 - could use a reference
Line 107- "Tadpoles were manually externally fertilized" - better say eggs or oocytes, than tadpoles
Line 166 - Supplementary table 1 doesn’t exist, is it figure s5? Some primers missing mmp1 et mmp13
Line 172, 173 - typos - biological significance
Line 158 – this information is not entirely clear "Five individual tadpoles were used in each extraction and pooled to a total of 10 per group for cDNA synthesis" - Did you pool 5 tadpoles in one sample and 10 of these samples constituted one exposure group?
Line 135- "They were then were rinsed once ..." remove one "were"
Line 199- "Interesting, when observing the remodeling trends of the individual ratio components, there seems to be overall better outcomes for lengthening rather than". -Can you reformulate and define what means “better”
Line 390 - correct "genes gene associated with..."
Line 394 -correct "Two prolactin mediated matric metalloproteinases, mmp1and 394 mmp13, were also observed to upregulated"
Figure S3- typo- "normalized" intensity in y axis legend
Question for the authors: Do you believe there is a potential threat to aquatic wildlife from environmental exposure to ivermectin residues?
This study offers novel insights into the physical and molecular characteristics of remodeling versus static, uncorrected defects, which is crucial for understanding the need for pre-metamorphic correction in vertebrates. Therefore, I recommend its publication following minor revisions.
Reviewer 3 Report
Comments and Suggestions for Authors
This is a well-designed study that characterized the cranio facial defects of pre-metamorphic Xenopus laevis tadpoles, in specific, cartilages, musculature and nerves. Authors treated tadpoles with either Ivermectin, a non-remodeling or Thioridazine, a remodeling model and assessed the craniofacial improvement. They further looked at qPCR of specific genes to relate the cranio facial remodeling. Overall, it is well designed study with following comments and suggestions from my side.
- Introduction is well written, however it’s hard to distinguish the specific things done in this particular study vs the background. For example, authors start mentioning 3 major groups of their study starting line 55. Then again gives background about 3 major contributors of cranio facial development. I think all the relevant background is important, but if the authors can first give all the necessary background and then specifically talk about the discoveries of the current study that gives better flow to the introduction.
- Line 54, if authors are talking about their previous work, please add reference.
- Line 77, “Although the ability to remodel some malformed structures has been previously described”, please add reference here.
- Line 209, Add “,” after stage 45
- Line 243- Please add specific model name (Ivermectin or Thioridazine) before “It is worth noting…” just to be specific.
- Line 341 – Add “to” before “compared”
- Line 385 – Please mention specifically RT-qPCR used to determine prl.2 and other gene expression.
- Line 390 – Remove “gene” after “genes”.
- Line 400 – I think it’s "though" not "thought" after “considered biologically significant,”
- Line 411 - Please start with brief introduction to your study along with unique questions asked and findings observed before discussing the results.
- Line 412 – “Gene expression profiles for our non-remodeling treatment show a likely role for the hormone, prolactin, in tissue remodeling rather than a more traditional metamorphic, thyroid hormone, mode of control”.
From the qPCR results, it looks like there is role of prolactin and mmps in the Thioridazine remodeling treatment model, but not Ivermentin non-remodeling treatment model. But line 412 shows prolactin role in non-remodeling treatment (Ivermectin)?
- Line 429, please add Thioridazone before “remodeling”
- qPCR shows only mRNA, Can the authors confirm the results with protein expression? With immunofluorescence, potentially authors could localize the increase or decrease in the respective protein expressions spatially to cartilage, muscle or peripheral neurons. The IF results may further add to the cranio facial remodeling observations following Ivermectin or Thioridazine treatments.
Round 2
Reviewer 3 Report
Comments and Suggestions for Authors
No further suggestions from my side.
Author Response
Reviewer 3: No further suggestions from my side.
Jones et al., : The authors thank the reviewer for their time.